# An Artificial Intelligence-Enabled ECG Algorithm for the Prediction and Localization of Angiography-Proven Coronary Artery Disease

**DOI:** 10.3390/biomedicines10020394

**Published:** 2022-02-07

**Authors:** Pang-Shuo Huang, Yu-Heng Tseng, Chin-Feng Tsai, Jien-Jiun Chen, Shao-Chi Yang, Fu-Chun Chiu, Zheng-Wei Chen, Juey-Jen Hwang, Eric Y. Chuang, Yi-Chih Wang, Chia-Ti Tsai

**Affiliations:** 1Division of Cardiology, Department of Internal Medicine, National Taiwan University Hospital Yun-Lin Branch, Yunlin County 640, Taiwan; y03687@ms1.ylh.gov.tw (P.-S.H.); y00906@ms1.ylh.gov.tw (J.-J.C.); y03065@ms1.ylh.gov.tw (S.-C.Y.); y00913@ms1.ylh.gov.tw (F.-C.C.); y03749@ms1.ylh.gov.tw (Z.-W.C.); 2Cardiovascular Center, National Taiwan University Hospital, Taipei 100, Taiwan; 3Graduated Institute of Biomedical Electronics and Bioinformatics, National Taiwan University, Taipei 106, Taiwan; r08945056@ntu.edu.tw; 4Division of Cardiology, Department of Internal Medicine, Chung Shan Medical University Hospital, Taichung 402, Taiwan; csy230@csmu.edu.tw; 5School of Medicine, Chung Shan Medical University, Taichung 401, Taiwan; 6Division of Cardiology, Department of Internal Medicine, National Taiwan University Hospital, Taipei 100, Taiwan; 7Bioinformatics and Biostatistics Core, Center of Genomic and Precision Medicine, National Taiwan University, Taipei 100, Taiwan

**Keywords:** artificial intelligence, deep learning, convolutional neural network, coronary artery disease

## Abstract

(1) Background: The role of using artificial intelligence (AI) with electrocardiograms (ECGs) for the diagnosis of significant coronary artery disease (CAD) is unknown. We first tested the hypothesis that using AI to read ECG could identify significant CAD and determine which vessel was obstructed. (2) Methods: We collected ECG data from a multi-center retrospective cohort with patients of significant CAD documented by invasive coronary angiography and control patients in Taiwan from 1 January 2018 to 31 December 2020. (3) Results: We trained convolutional neural networks (CNN) models to identify patients with significant CAD (>70% stenosis), using the 12,954 ECG from 2303 patients with CAD and 2090 ECG from 1053 patients without CAD. The Marco-average area under the ROC curve (AUC) for detecting CAD was 0.869 for image input CNN model. For detecting individual coronary artery obstruction, the AUC was 0.885 for left anterior descending artery, 0.776 for right coronary artery, and 0.816 for left circumflex artery obstruction, and 1.0 for no coronary artery obstruction. Marco-average AUC increased up to 0.973 if ECG had features of myocardial ischemia. (4) Conclusions: We for the first time show that using the AI-enhanced CNN model to read standard 12-lead ECG permits ECG to serve as a powerful screening tool to identify significant CAD and localize the coronary obstruction. It could be easily implemented in health check-ups with asymptomatic patients and identifying high-risk patients for future coronary events.

## 1. Introduction

Coronary artery disease (CAD) is one of the major cardiovascular diseases globally. It has a high impact on long-term mortality and morbidity in both developed and developing countries [1]. The previous study had shown that ischemic heart disease leads to around 16% cause of death among all causes [2]. Epidemiology survey also revealed the increasing incidence of CAD globally. However, the screen and assessment of CAD depend on classical symptoms, signs, and other comorbidities.

For further evaluation of possible CAD, there are many non-invasive test modalities, such as stress ECG test, nuclear medicine-based cardiac perfusion scan, single-photon emission computed tomography (SPECT), cardiac magnetic resonance (CMR), and computed tomography angiography (CTA) [3]. However, these tests generally are not easily accessible, require specialized equipment, and are also time-consuming and expensive. The sensitivity and specificity of these tests are not optimal (around 75–90%) [4]. There are also concerns about radiation exposure in nuclear medicine or computed tomography-based exams. Furthermore, stress-based tests that require patient exercise are sometimes not doable in debilitating patients. Accordingly, we need a test that is easily accessible, available, cheap, and highly accurate to predict CAD.

ECG is a non-invasive test. It has the advantages of easy to use, reproducible, widely available, and also inexpensive, compared to previously mentioned test modalities. ECG can be used to detect significant CAD by significant ECG changes, such as ST-segment deviation, T-wave inversion, and Q-wave appearance [5,6]. However, the interpretation accuracy was affected by other conditions, such as arrhythmia, cardiomyopathy, bundle branch block, early repolarization, etc. [7]. 

Artificial intelligence (AI) in the form of deep learning convolutional neural networks (CNNs) had been used in many disease models before [8]. CNN is a model that can be trained by inputting a large amount of data to automatically extract meaningful features from one-dimensional (e.g., signal) or two-dimensional (e.g., image) data and perform classification or regression tasks. In the medical field, both signal and image data are helpful for clinical diagnosis [9,10]. The AI-enabled ECG (AI ECG) algorithm with deep learning extracts uses meaningful patterns of complex information from ECG. It has been demonstrated to be effective in identified patients with heart failure, left ventricular hypertrophy, paroxysmal atrial fibrillation, and other cardiovascular diseases [11,12]. However, using it to identify patients with significant CAD has never been studied.

We hypothesized that the application of an AI ECG might provide an efficient way for identifying patients with CAD. To test this hypothesis, we performed a retrospective study of comparing ECG between patients with invasive coronary angiography documented significant CAD and normal people to develop and validate a deep learning AI model for detecting CAD and even predicting the location of the obstructed vessel using standard 12-lead ECG.

## 2. Materials and Methods

### 2.1. Study Populations

This was a multi-center retrospective cohort study with patients with angiography-proven significant CAD receiving invasive coronary angiography with percutaneous coronary intervention in the National Taiwan University Hospital and National Taiwan University Yun-Lin Branch from 1 January 2018 to 31 December 2020. All the patients received coronary percutaneous coronary intervention (PCI) with stent implant according to Taiwan’s National Health Insurance policy as severe stenosis more than 70% by quantitative coronary angiography assessment or intermediate lesion defined as coronary stenosis between 50–70% with physiological significance (coronary fractional flow reserve < 0.80). The control patients are those in the cardiovascular clinics with neither documented CAD nor positive stress test. We excluded patients who used drug-coated balloons to treat the stenosis to increase the uniformity of the case group. We also defined non-CAD patients as having no prior cardiac history at the time of data collection, similar to previous literature [13]. All patients were followed by their physicians in cardiology out-patient clinics. The study was approved by the Institutional Review Board of the National Taiwan University Hospital (No. 202101049RINC), which waived informed consent.

### 2.2. Data Collection and Parsing

The algorithm of ECG collection and parsing is shown in Figure 1. A total of 2303 patients with angiography-proven significant CAD were included and a total of 12,954 records of 12-lead ECG were collected. The parsing criteria was based on less than 30 days before the date of the patient’s PCI record. During this period, all 12-lead ECGs of these patients were included in our dataset. After parsing by the criteria, 1635 CAD patients met the criteria and a total of 3221 ECG records (labeled as CAD) were collected. Then, according to the patient’s catheterization results with stent implant, we divided them into three groups: stenosis and stent implant in the left anterior descending artery (LAD), left circumflex artery (LCX), and right coronary artery (RCA).

Among them, there were 899 patients in the LAD group, with 1666 ECG records, 228 patients in the LCX group, with 425 ECG records, and 508 patients in the RCA groups, with 1130 ECG records. There were 1053 normal patients without CAD in the control group, with 2090 ECG records (labeled as NOR).

### 2.3. Dataset Preparation and Data Analysis

Our research is to use CNN model and the 12-lead ECG of CAD patients to predict which vessel is obstructed. In addition to the three types of coronary artery (LAD, LCX, and RCA), we also added normal patient’s ECG, who showed no evidence of CAD (labeled as NOR) to distinguish the CAD and non-CAD patients.

In our experiment, the dataset was divided into a training set, validation set, and test set, and the split ratio is as follows: [Train/Validation/Test] = [0.85/0.05/0.10]. It should be noted that there is an imbalance in the amount of data between the groups. The imbalance between the groups was found in a pioneer study to cause bias during model training. Therefore, we used a down-sampling strategy to randomly select data to form a subset to achieve a balance of ECG records between all groups. Moreover, we also found the data of the same patient could not be included both in the training and validation datasets; otherwise, it might affect the credibility of the results.

While training with down-sampling subsets, we found that the model has a higher classification ability for ECGs with acute myocardial infarction (AMI) or ischemia labeled in the ECG diagnosis. Therefore, in addition to randomly selecting three class-balanced subsets, there are also two class-balanced subsets selected as subgroup datasets. According to the ECG diagnostic information, the ECG data with AMI or ischemia were selected to form a subset (called “subgroup (1)”). The less discernible data were selected to form the other subset (called “subgroup (2)”) (Appendix A). Then, we trained both subsets separately to verify our findings.

### 2.4. Data Type and Preprocessing

There are two types of 12-lead ECG records: one-dimensional time series data and two-dimensional images. Most research was conducted using one-dimensional time series data. In recent years, due to the breakthrough of CNN in image recognition, papers using 12-lead ECG images for research have been published. The data collected from the hospitals in this study was image data, which were cropped from the patient’s ECG records. This image is a standard 12-lead ECG image, including lead I, II, III, V1~6, aVR, aVL, and aVF, produced by MAC 3500, GE Healthcare (Chicago, IL, USA). Its measurement frequency is 500 Hz and the measurement of each lead duration is 2.5 s. Before starting the training, the red grid background was first removed to make the whole picture more precise and focus on the ECG signal itself. The methods of background grid removal and cropping are shown in Appendix A. After pre-processing, the size of the image will be adjusted to 512 × 256 × 3, and following CNN model training, it will also be trained by this size.

### 2.5. Model Build-Up

According to the dimensionality of the data in the computing domain, the image is two-dimensional (2D).

In CNN model construction, we used various models and their parameters and have achieved promising results in the ImageNet image recognition competition in the CNN part to do transfer learning. A total of six well-known network architectures (VGG16 [14,15], ResNet50V2 [16], InceptionV3 [17], InceptionResNetV2 [18], Xception [19], and DenseNet [20,21]) had been tried. After the features were extracted by CNN from the 12-lead ECG image, the features were flattened by GlobalAveragePooling (GAP) [22]. GAP is a pooling operation that aims to flatten the feature maps of the last convolutional layer into a one-dimensional feature vector by averaging the values of each feature map. After obtaining the feature vectors, we tried the connection with and without an intermediate dense layer (that is Dense 1 shown in Figure 2) to see which architecture performing better. Next, we added Dropout (drop rate = 0.5), which randomly sets the output features to 0 with the frequency of rate at each step during the training process, to avoid overfitting. Finally, taking the stage two classifier for example, we added another dense layer with a size of four, representing the four types of LAD, LCX, RCA, and NOR, as the output layer. The image input model architecture was shown in Figure 2.

### 2.6. Training Process

The training platform used in this study is Google Colaboratory (Colab) [23] with a high-RAM GUP environment. Colab is a cloud computing environment that supports python 3.8 and Tensorflow package, which is commonly used to build and train CNN. As a Google resource, Colab can be linked with Google Drive, which allows users to access the files in Colab by uploading datasets to personal Google Drives.

We use the Keras application programming interface (API) to build CNN models. In addition to allowing users to easily build a CNN model, the keras API also provides a simple package that allows users to use various well-known models that have won awards in the ImageNet competition to do transfer learning. In addition to model building, keras also provides various APIs and functions that can be used, such as callbacks, optimizers, metrics, losses, etc. Appendix A shows the settings of the APIs and the training parameters in this study.

### 2.7. Evaluation Metrics

The primary outcome of the study was the ability of the AI-enhanced ECG to identify patients with angiography-proven significant CAD using a standard 12-lead ECG recorded at baseline. This performance was assessed by several evaluation metrics, such as the area under the curve (AUC) of the receiver operating characteristic (ROC) curve, accuracy, precision, and recall. All metrics are presented by taking the mean of five times of repetitive training and present with the mean and standard deviation with 95% confidence interval.

The confusion matrix defined four important terms, True Positive (TP), False Positive (FP), Ture Negative (TN), and False Negative (FN), which were used to calculate the metrics we mention above. Accuracy is a metric that evaluates the classification ability of the models by calculating the proportion of samples that are correctly classified among all samples (as Formula (1) shown). Precision indicates the percentage of predicted positive results that are true positives. It is calculated by dividing the number of true positive samples by the sum of true positive and false positive samples (shown in Formula (2)). Recall is a more meaningful metric that reveals the percentage of predicted positive samples that are correctly classified. It is calculated by dividing the number of true positive samples by the sum of true positive and false negative samples (shown in Formula (3)):(1)$$Accuracy=\frac{TP+TN}{TP+FP+TN+FN}$$
(2)$$Precision=\frac{TP}{TP+FP}$$
(3)$$Recall=\frac{TP}{TP+FN}$$

ROC is defined as the changes between the true positive rate (TPR), known as sensitivity, and the false positive rate (FPR), known as 1-specificity, under various decision thresholds. The ROC curve graphically combines sensitivity and specificity to accurately reflect the relationship between the specificity and sensitivity of the testing result. By shifting the threshold from 0 to 1, we can obtain a series of coordinates from TPR and FPR. With FPR as the X-axis and TPR as the Y-axis, the ROC curve can be plotted.

AUC is the ratio of the area covered under the curve (here, we used the ROC curve), divided by the total area. Calculating the AUC allows us to quantify the ROC curve and use it to compare the performance of the models. The model can be graded into four levels by using AUC, described as follows:(1)AUC < 0.5 (no discrimination)(2)0.7 ≤ AUC < 0.8 (acceptable discrimination)(3)0.8 ≤ AUC < 0.9 (excellent discrimination)(4)0.9 ≤ AUC ≤ 1.0 (outstanding discrimination)

The ROC-AUC analysis is used in the fields of radio, biology, criminal psychology, and more recently in machine learning and data mining [24]. In medicine, it is widely used in the diagnosis of diseases, but also in epidemiology, empirical medical research, radiological techniques, and social science research [24]. The advantage of the ROC curve is that it is straightforward to observe the clinical accuracy of the analysis method through the graphical representation.

## 3. Results

### 3.1. Image Input Model Architecture Optimization

To optimize image model architecture, we used the subgroup (1) dataset to test the different pre-trained CNN models connected with and without the dense (fully-connected) layer and choose the best model with the highest accuracy as the core CNN model. The experimental results are shown in Table 1. As Table 1 shows, the InceptionV3 CNN model without the dense layer was the best architecture with the highest accuracy of 0.9. Therefore, the rest of the datasets, including randomly selected and subgroup (2) datasets, were also trained with this architecture.

### 3.2. Detection of CAD and Prediction of the Obstructed Coronary Vessel

#### 3.2.1. Random Selection Dataset

To eliminate the interference of class imbalance, we did randomly select down-sampling in the LAD and RCA categories to balance the amount of ECG data in all categories. To ensure the credibility of the experiment, we did repetitive training and randomly selected a total of three different subsets to calculate the average test score. The result is shown in Appendix A.

The Appendix A shows the result of the image input model. For the image input model, the AUC of each category was NOR (1.0), LAD (0.885), LCX (0.776), and RCA (0.816). The corresponding confusion matrix and ROC curve are shown in Figure 3a,d.

When analyzing the performance of each subset in both input models, we found that subset 3 had the highest accuracy among all three subsets. To know the key factor why subset 3 can get higher accuracy, we did the comparison of these three subsets. We found that there were more ECG with features or diagnosis of myocardial ischemia in subset 3 compared to the rest of the two subsets, which might cause performance differences among the subsets.

#### 3.2.2. Subgroup Datasets

To verify that features of AMI and ischemia have a high correlation for diagnosing CAD, we then proceed with the subgroup analysis. The first subgroup had those ECGs with AMI or ischemia in the diagnosis, and the second had those without AMI or ischemia in the diagnosis. Table 2 shows the subgroup test scores. Table 2 shows the results of the image input model.

In subgroup (1), as we mentioned in the method, the dataset was down-sampled by certain criteria, where we only included ECGs with AMI or ischemia into the dataset. In the image input model, the accuracy reached 0.9, and the AUC of all categories was greater than or close to 0.95. The corresponding confusion matrix and ROC curve are shown Figure 3b,e. On the other hand, subgroup (2) was associated with a low model performance and the model cannot distinguish LAD, LCX, and RCA well. It could only effectively distinguish between NOR and CAD. The corresponding confusion matrix and ROC curve are shown in Figure 3.

Finally, we provided the evidence of capability of our AI model to predict the coronary lesions. In Figure 4, we show three ECG examples that could not be used to determine the coronary lesions and their angiography results. The AI successfully determined the location of coronary obstruction and the angiography showed the presence of more than 70% stenosis in the AI-predicted coronary lesions.

## 4. Discussion

CAD and its complications pose a high socioeconomic burden and a high impact on the overall survival of society [25]. According to current guidelines, the detection of significant CAD relies on costly and time-consuming examinations, such as SPECT, CMR, CTA, etc. (sensitivity 57–90%, specificity 60–94%) [3,26]. ECG is a widely available and not expensive examination. It is commonly used as the golden diagnostic study for detecting AMI. However, its efficacy and accuracy for detecting myocardial ischemia or stable CAD is not as good as its ability to predict AMI.

In this study, we found that an AI-enable ECG algorithm can be an effective tool for detecting significant CAD from baseline 12 lead ECG (Marco-average AUC 0.869 for the image input model). Our study provides one alternative timely efficient and cost-effective tool for predicting significant CAD by the AI-enhanced ECG algorithm in common clinical settings. Furthermore, our study also shows that an AI-enhanced ECG algorithm could determine which coronary artery is stenotic or obstructed when the ECG performs the symptom of AMI and ischemia.

### 4.1. Importance of Model Optimization

Our AI model is better than the previous method based on simple feature classification. It may be because the machine can recognize more embedded features which we would not have recognized in the past if we had chosen an appropriate model. In our research, different CNN architectures had a great impact on the accuracy of classification.

Taking the image input model as an example, as shown in Table 1, all the CNN models without connecting to a dense layer got better results, except for VGG16. We assume that the features that had been taken out from the architecture of these CNN models are good enough; hence, there is no longer a need to add another intermediate layer to adjust the feature weights. It is better to connect directly to the output layer for prediction.

However, VGG16 presented the opposite results. When we removed the dense layer from the fully-connected layer of VGG16, the accuracy was only 0.25, which means the model was not doing the classification. The problem would be caused by the downsampling process in the Maxpooling layer, since the VGG16 architecture performs Maxpooling many times, which in turn leads to information deprivation and loss of discriminative capability. On the contrary, other architectures use the concept of a residual block—the building block of ResNet (12)—to preserve the upper-level features, so the feature extraction capability is relatively better.

### 4.2. AI in Significant CAD Detection

Our study also shows that if the ECG has some unique features, the AI-based model performs well. In our subgroup analysis, the AI model has a Marco-average AUC of more than 0.95 to predict which coronary artery has significant stenosis, i.e., more than 70%. This finding is compatible with the basics of image-based AI learning because patterns or features recognition is the key to establishing a highly accurate AI model [27]. Another advantage of our study is that we used invasive angiography to define the presence of significant coronary stenosis, which is the gold standard to diagnose CAD. In most of the previous studies of AI-based ECG prediction of CAD, CAD was not diagnosed by the invasive coronary angiography, but by open-source datasets or myocardial infarction patients, and even only using lead II ECG for interpretation [28,29,30]. Therefore, we first demonstrate that AI-based ECG pattern recognition could predict true coronary disease or obstruction and clearly discriminate which coronary vessel is stenotic or severely obstructed if an appropriate training model is incorporated.

This study is very important because it provides a simple way to identify individuals at high risk of significant CAD from the general population, who benefit from invasive coronary angiography with stent implant, and maybe further improving long-term prognosis and even prevent sudden cardiac death [3]. AI-based ECG testing to identify significant CAD is very efficient and highly cost-effective, especially in the settings of routine health exams, in which many patients are asymptomatic.

There are limitations in the present study. The main purpose in this study was to identify patients with documented CAD from the normal people, and this study sought to distinguish those people with significant CAD from the normal people. In normal people, not all the people received angiography. Therefore, some normal people may have severe CAD and silent ischemia, and this is the inherent limitation of choosing controls without angiography. However, our results showed that the performance of using this kind of control patient was acceptable. Nevertheless, our AI algorithm still carries the risk of misclassification of patients with CAD as normal people, which, thus, delays their treatment. Because no non-invasive examination is 100% accurate, we still hope an AI-enabled method for detecting significant CAD can become one of the evaluation tools for detecting and localizing CAD, as it has the potential to be quick and simple, not as time-consuming and expensive, and less harmful than radiation-exposing screening tools, such as stress nuclear medicine or coronary computed tomography.

## 5. Conclusions

In this paper, we validated the feasibility of using the 12-lead ECG and deep learning AI model to detect the most accurate angiography documented CAD. AI-based ECG could even determine which coronary artery is stenotic or obstructed from image data derived from ECG if a proper AI learning model is chosen.

## Figures and Tables

**Figure 1 biomedicines-10-00394-f001:**
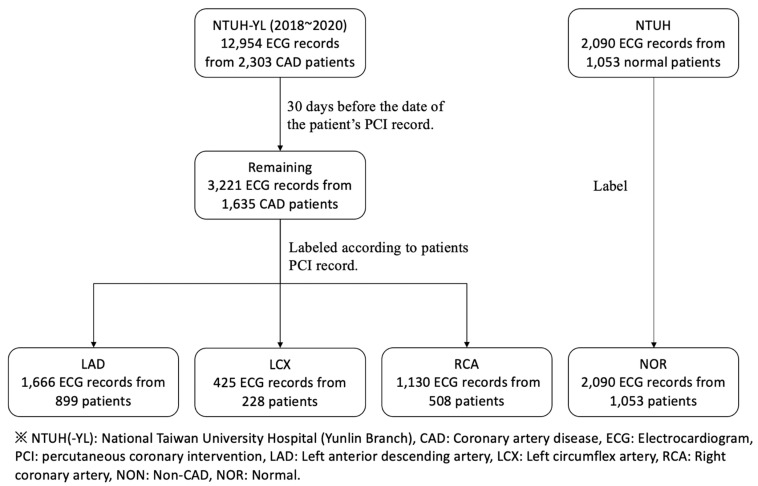
Data collection and parsing protocol.

**Figure 2 biomedicines-10-00394-f002:**
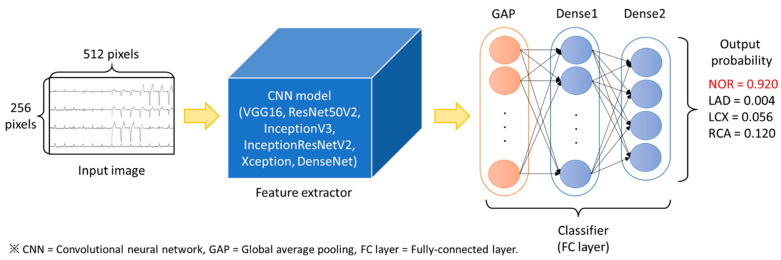
Model architecture of the 2D CNN model (output probability number is only for illustration).

**Figure 3 biomedicines-10-00394-f003:**
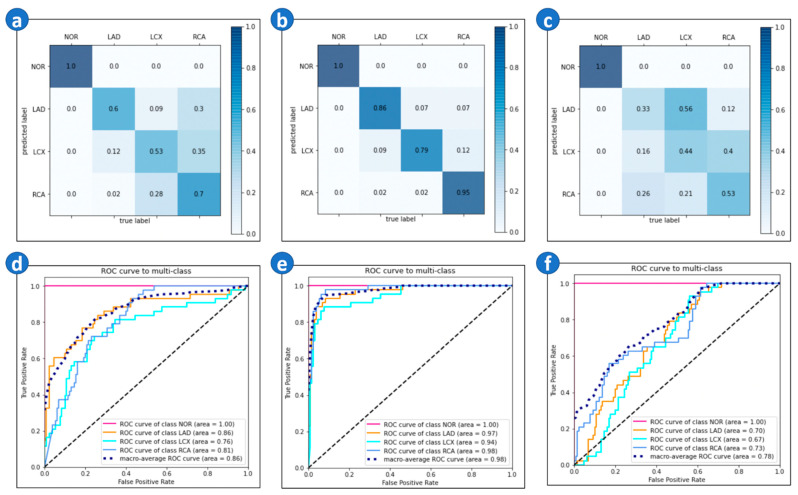
Confusion matrix and ROC curve. (**a**) Confusion matrix of the image input model of random selection. (**b**) Confusion matrix of the image input model of subgroup (1). (**c**) Confusion matrix of the image input model of subgroup (2). (**d**) ROC curve of the image input model of random selection. (**e**) ROC curve of the image input model of subgroup (1). (**f**) ROC curve of the image input model of subgroup (2).

**Figure 4 biomedicines-10-00394-f004:**
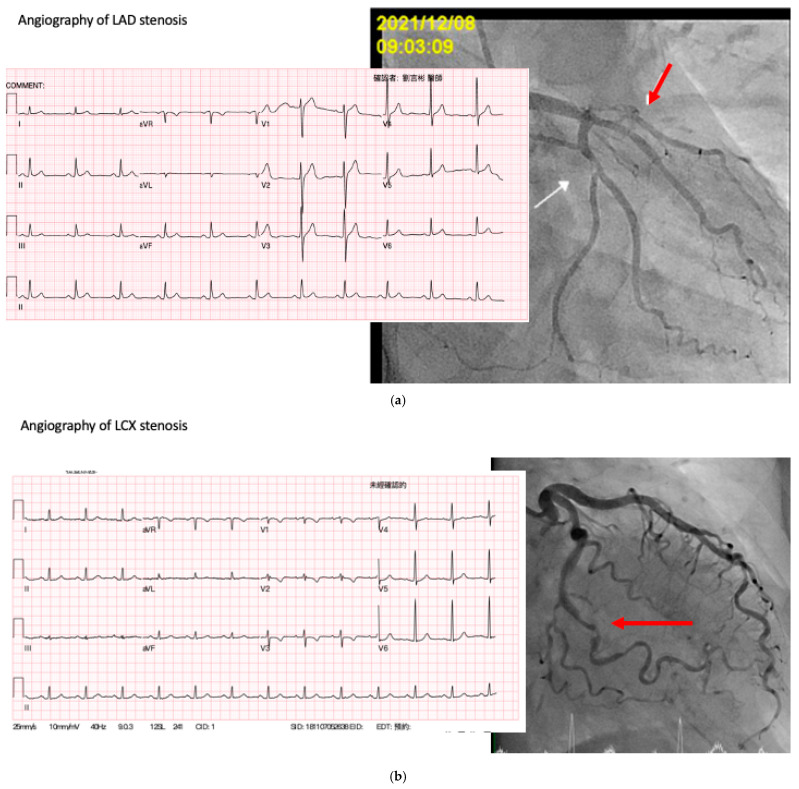
ECG examples of the AI model predicting the coronary lesions, arrows indicate stenosis by coronary angiography result. (**a**) ECG of a patient with >70% stenosis in LAD and about 30% stenosis in LCX in angiography. (**b**) ECG of a patient with >70% stenosis in LCX in angiography. (**c**) ECG of a patient with >70% stenosis in RCA in angiography.

**Table 1 biomedicines-10-00394-t001:** Image input model architecture optimization—evaluation metrics with mean and standard deviation with 95% confidence interval.

w/							
Model	Acc.	AUC				Precision	Recall
		NOR	LAD	LCX	RCA		
VGG16	0.670 ± 0.030	1.000 ± 0.000	0.873 ± 0.036	0.807 ± 0.064	0.913 ± 0.024	0.720 ± 0.039	0.674 ± 0.020
ResNet50V2	0.827 ± 0.007	1.000 ± 0.000	0.927 ± 0.026	0.903 ± 0.024	0.950 ± 0.000	0.836 ± 0.012	0.831 ± 0.12
Xception	0.850 ± 0.023	1.000 ± 0.000	0.940 ± 0.023	0.907 ± 0.033	0.963 ± 0.007	0.855 ± 0.009	0.851 ± 0.011
Inception ResNetV2	0.857 ± 0.035	1.000 ± 0.000	0.957 ± 0.013	0.943 ± 0.017	0.96 ± 0.011	0.847 ± 0.012	0.840 ± 0.012
DenseNet121	0.843 ± 0.007	1.000 ± 0.000	0.953 ± 0.007	0.920 ± 0.023	0.953 ± 0.007	0.851 ± 0.014	0.831 ± 0.012
InceptionV3	0.876 ± 0.025	1.000 ± 0.000	0.958 ± 0.019	0.944 ± 0.024	0.970 ± 0.011	0879 ± 0.020	0.873 ± 0.025
w/o							
VGG16	0.250 ± 0.000	0.500 ± 0.000	0.500 ± 0.000	0.500 ± 0.000	0.500 ± 0.000	0.085 ± 0.043	0.252 ± 0.004
ResNet50V2	0.854 ± 0.013	1.000 ± 0.000	0.952 ± 0.004	0.908 ± 0.010	0.966 ± 0.005	0.856 ± 0.017	0.852 ± 0.015
Xception	0.856 ± 0.005	1.000 ± 0.000	0.954 ± 0.021	0.928 ± 0.016	0.968 ± 0.004	0.857 ± 0.006	0.855 ± 0.005
Inception ResNetV2	0.872 ± 0.010	1.000 ± 0.000	0.950 ± 0.015	0.924 ± 0.017	0.976 ± 0.010	0.875 ± 0.009	0.872 ± 0.010
DenseNet121	0.890 ± 0.014	1.000 ± 0.000	0.978 ± 0.007	0.936 ± 0.025	0.966 ± 0.012	0.893 ± 0.010	0.889 ± 0.013
**InceptionV3**	**0.900 ± 0.012**	**1.000 ± 0.000**	**0.966 ± 0.010**	**0.948 ± 0.014**	**0.978 ± 0.010**	**0.903 ± 0.011**	**0.899 ± 0.012**

Acc.: Accuracy, AUC: The area under the ROC curve, NOR: Normal, LAD: Left anterior descending, LCX: Left circumflex artery, RCA: Right coronary artery, w/: with a dense layer, w/o: without a dense layer.

**Table 2 biomedicines-10-00394-t002:** Subgroup datasets scores—evaluation metrics with mean and standard deviation with 95% confidence interval (image input model).

Image Input Model							
Subgroup	Accuracy	AUC				Precision	Recall
		NOR	LAD	LCX	RCA		
**1**	0.973 ± 0.012	1.0 ± 0.0	0.966 ± 0.010	0.948 ± 0.014	0.978 ± 0.010	0.903 ± 0.011	0.899 ± 0.012
**2**	0.566 ± 0.008	1.0 ± 0.0	0.710 ± 0.040	0.672 ± 0.029	0.704 ± 0.040	0.553 ± 0.045	0.563 ± 0.006

AUC: The area under curve, LAD: Left anterior descending, LCX: Left circumflex, RCA: Right coronary artery.

## Data Availability

The data that support the findings of this study are available from the corresponding author, C.T. Tsai, upon reasonable request.

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
