# Peer review of "An Artificial Intelligence-Enabled ECG Algorithm for the Prediction and Localization of Angiography-Proven Coronary Artery Disease"

_biomedicines, 2022, doi:10.3390/biomedicines10020394_

Round 1

Reviewer 1 Report

The authors have accessed a valuable database for this study, a large number of ECGs associated with coronary angiography results.

While the report evidently reflects considerable effort on the part of the authors, unfortunately it is seriously flawed. The "control" consists of patients not assessed by coronary angiography. We don't know whether they have CAD, only that through normal, but undocumented, clinical judgement, it was deemed sufficiently unlikely that angiography was not ordered. Silent ischemia is sufficiently common that an unknown portion may be false, rather than true, negatives, rendering this population unusable as a control. Moreover, we know little about the patient populations in the various analysis groups, such as their demographics and medical histories. And there is no account of patients with negative angiograms, their characteristics or their cardiograms. There is no accounting for the instability of CAD, multiple ECGs being accepted from patients without regard to changes among them within patients. Analytic treatment of patients with documented multi-vessel disease, which is common, is unexplained.

The extensive efforts expended to reconstruct time series data from images was misguided, as rendering images from the raw time series data that is the basis for the extensive history of advances in electrocardiographic research and development inevitably irreversibly discards the vast majority of the information in the signals. It is unsurprising that this data supported poorer results than the images themselves.

The report is excessively vague in important places. Cases are sorted according to whether they show signs of infarction or ischemia that are undefined, trivializing the hundreds of interpretative statements and confidence assessments in modern computerized diagnostic cardiography analyses. A long list of alternative CNN architectures with many options in structure are introduced without description or documentation. Entire elements (e.g. GAP) and processes (e.g. stride) are unexplained. Important modifications are described as "imitating" alternatives, without description or justification. Ultimately, it is not clear which designs are included with which options and features. The results of multiple tests are combined without statistical rigor (standard deviations and 95% confidence limits are related, but different concepts - it is not clear what the expression "standard deviation of 95% confidence level" means, or what its statistical legitimacy might be). What documentation is provided about the CNN architectures is largely unsatisfactory, relying on dated pre-print server records and other non-reviewed reports. Pre-print servers are increasingly popular, but they are not peer-reviewed science, and are unacceptable as documentation in a peer-reviewed report, which this study aspires to be. The role of such services is rapid publication of preliminary reports prior to peer-review. Reports of satisfactory studies should be replaced by peer-reviewed manuscripts in short order, but many of the citations used herein are 5 or more years old. Ultimately, while the authors dismiss 50 years of research and development of computerized electrocardiography analysis by reference to a review article in an obscure journal, what they propose to have established is that with CNNs they can almost replicate the performance of those algorithms. Cardiologists are historically skeptical of neural network analysis because its decision process cannot be explained. They will have no motivation to defer to it unless proof of superiority is provided. There are tantalizing hints of possibilities herein, such as localization of culprit arteries (though this is not unprecedented in computerized diagnosis based on heuristics) but their derivation is not sound.

Author Response

Thanks for your important comments. 

I had replied in the attaching file and also revised the manuscript accordingly.

Reviewer 2 Report

In this study the Authors have used CNN method to analyze the results of ECG in order to determine whether it is possible to used this AI method in the diagnosis of CAD. While the study is interesting and the results are convincing, it also requires some revision.

My major concern is why the Authors have decided to use CNN instead of time delay neural network (TDNN).

The quality of Figure 1 must be improved.

In the introduction the CAD is nicely and comprehensively described. However, the method used in this study (CNN) is only briefly mentioned. There are multiple studies and some nice reviews of applications of CNN in the prediction of diseases. The Authors should mention at least some of them.

Figure 2 suggests that the output probability was the same, regardless of the chosen architecture.

Table 1, why is “InceptionV3” bolded?

Line 252, to be precise this improvement of accuracy was over 12.8 % and not almost 10%, there is a difference between percentage and percentage point.

Lines 340-348, this is obvious and true for all of the AI methods. I suggest to remove this part.

Line 350, some values must be provided and compared to support this statement.

Lines 411-412, supplementary materials must be listed here

Author Response

(The authors gave the same response as above.)

Round 2

Reviewer 1 Report

The authors have improved wording in a couple of instance (though not all - e.g. "Marco-average" is not a thing), but the structural flaws in this study:

  • controls declared as "stable non-cardiac patients with low cardiovascular risk factors", which is inconsistent with some, unstated, portion of them being subjected to stress tests and echocardiograms, and characterized as CAD negative without direct evidence
  • lack of specificity about ECG diagnostics
  • conflation of ECG signs of ischemia and if infarction, which have different implications for CAD
  • retention of pointless serialization of time series from images, despite inevitable destruction of information in rendition of the images

as well as continued extensive reliance on citation of non-peer reviewed sources make this manuscript unsuitable for publication in a peer reviewed journal.

Author Response

Thanks for these important comments. 

We all revised and corrected as your suggestions.

Thanks again. 

Reviewer 2 Report

The Authors have corrected the manuscript. Current version can be accepted.

Author Response

Thanks for your important comment.
